# A Durable Hybrid RAM Disk with a Rapid Resilience for Sustainable IoT Devices

**DOI:** 10.3390/s20082159

**Published:** 2020-04-11

**Authors:** Sung Hoon Baek, Ki-Woong Park

**Affiliations:** 1Department of Computer System Engineering, Jungwon University, Chungcheongbuk-do 28024, Korea; shbaek@jwu.ac.kr; 2Department of Computer and Information Security, Sejong University, Seoul 05006, Korea

**Keywords:** IoT, sustainability, hybrid RAM disk, direct byte read, secondary storage, operating system

## Abstract

Flash-based storage is considered to be a de facto storage module for sustainable Internet of things (IoT) platforms under a harsh environment due to its relatively fast speed and operational stability compared to disk storage. Although their performance is considerably faster than disk-based mechanical storage devices, the read and write latency still could not catch up with that of Random-access memory (RAM). Therefore, RAM could be used as storage devices or systems for time-critical IoT applications. Despite such advantages of RAM, a RAM-based storage system has limitations in its use for sustainable IoT devices due to its nature of volatile storage. As a remedy to this problem, this paper presents a durable hybrid RAM disk enhanced with a new read interface. The proposed durable hybrid RAM disk is designed for sustainable IoT devices that require not only high read/write performance but also data durability. It includes two performance improvement schemes: rapid resilience with a fast initialization and direct byte read (DBR). The rapid resilience with a fast initialization shortens the long booting time required to initialize the durable hybrid RAM disk. The new read interface, DBR, enables the durable hybrid RAM disk to bypass the disk cache, which is an overhead in RAM-based storages. DBR performs byte–range I/O, whereas direct I/O requires block-range I/O; therefore, it provides a more efficient interface than direct I/O. The presented schemes and device were implemented in the Linux kernel. Experimental evaluations were performed using various benchmarks at the block level till the file level. In workloads where reads and writes were mixed, the durable hybrid RAM disk showed 15 times better performance than that of Solid-state drive (SSD) itself.

## 1. Introduction

A reliable but responsive storage device is an inevitable concern for realizing sustainable Internet of things (IoT) devices for mission-critical systems [1]. Unlike general consumer devices, sustainable IoT devices must perform their own tasks correctly in a stable manner without failures. As shown in Figure 1a, mission-critical systems need to ensure rapid system recovery and resilience, even in the face of a sudden power failure because one failure inside a mission-critical system may result in a mission failure or a tragedy. Therefore, realizing sustainable IoT devices ensuring low read latency for accessing critical data, as well as data durability for storing critical data, is a critical factor for their sustainability [2]. Short boot times are also an important factor in mission-critical IoT systems because it should be ready for certain mission-critical tasks even after a sudden power failure. Regarding these points, flash-based storage is considered to be a de facto storage module for sustainable IoT platforms under a harsh environment due to its relatively fast speed and operational stability compared to disk storage. To achieve these design goals, usually flash-based Solid-state drive (SSD) is installed with the configuration of single attached or Redundant array of inexpensive disks (RAID)manner, and mostly, IoT operating systems such as Android Things [3], and an I/O subsystem in the operating system, manages the storage device [4]. Although their performance is considerably faster than disk-based mechanical storage devices, the read and write latency still could not catch up with that of Random-access memory (RAM). Therefore, RAM could be used as a storage device or system for time-critical IoT applications.

A traditional storage I/O subsystem, even if it uses flash-based SSD, will have larger latency and lower bandwidth compared to memory-oriented read/write operations. Even though the performance gap diminishes, there still exists several orders of magnitude for latency. Additionally, RAM can access data on a byte-addressable level. There are a lot of studies to improve I/O performance with RAM memory, and most of them use RAM memory as a buffer for storage devices [5,6,7,8,9]. The read and write buffer can reduce write latency by buffering incoming data to a RAM buffer, or reduce read latency to get data directly from a RAM buffer, with appropriate buffer-management algorithms. However, the buffer-based approach still must go through existing file system operations and may have memory management overhead such as page cache.

On the other hand, the RAM disk is a software program that turns a portion of the main memory into a block device [10,11,12]. The RAM disk is the most expensive and fastest storage device, in which the RAM memory block device works like a disk drive. It is also referred to as a software RAM drive to differentiate it from a hardware RAM drive. RAM disks can provide fast I/O response and low latency when accessing data. However, it has the disadvantage of data loss in the event of a power failure; therefore, it lacks durability and persistency. To address this problem, many studies have been conducted on various types of systems. The simplest approach to prevent data loss is to asynchronously dump the entire contents of the RAM disk into a dedicated hard disk drive [13]. For better durability, dumping into a hard disk drive can be performed synchronously, but this method would use the inefficient traditional I/O for RAM-based storage [14]. In addition, there is little scope in terms of byte addressability advantage of RAM devices when designing RAM-based storage systems. The byte addressing data access operation provides very low latency.

Figure 1b,c illustrate a basic difference between a RAM disk and the proposed scheme, durable hybrid RAM disk (DHRD). The generic RAM disk has a block interface and loses data at a power failure. However, DHRD with direct byte read (DBR) can improve both durability and read throughput. The DHRD consists of a RAM disk and a non-volatile storage, such as SSD, as a hybrid storage system. With this hybrid approach, the DHRD provides durability, which means it does not suffer data loss during sudden power failure. The DBR DHRD performs I/O operations with the new read scheme that does not use a disk cache, i.e., page cache, for reads and is byte-addressable unlike direct I/O, wherein direct I/O uses a strict block-addressing mode. The byte-addressable feature is more convenient for applications than the use of direct I/O. The new byte-addressable read scheme can be mixed with buffered writes to apply it to the hybrid RAM disk; moreover, it can be applied to existing applications without any modification. In addition to that, the initialization procedure of DBR DHRD can reduce the boot time of the storage device, since it allows general I/O requests during the initialization process itself, while other RAM disk-based storage cannot support general I/Os during the initialization. Various experiments that could be applied to sustainable IoT devices were performed to DBR DHRD and other storage configurations such as SSD and hybrid storage device. The experimental results show that the DBR DHRD gives better I/O performance than others.

The rest of this paper is organized as follows: Section 2 provides an overview of related work, Section 3 presents the design and implementation of DHRD in detail. The performance evaluation of DHRD is presented in Section 4. Finally, Section 5 concludes this paper and presents the relevant future work.

## 2. Related Work

RAMDisk is a software-based storage device that takes a portion of the system memory and uses it as a disk drive with legacy file system operations. The more RAM your computer has, the larger the RAM disk you can create, but the cost would also be more.

Recently, RAM has been used as a storage system for several high-end computing systems and IoT devices to provide low latency and low I/O overhead. RAM is used for intensive random I/O in various fields, such as in-memory databases [15], large-scale caching systems [16], cloud computing [17,18,19,20], virtual desktop infrastructure [21,22,23], web search engine [24], and mission-critical systems like space applications [25].

Several RAM disk devices were previously developed such as [10,11,12]. The most traditional application of RAM disk modules is their use as virtual file systems for Linux kernel from the system’s boot time. During the system boot time, Linux kernel uses more than one RAM disk file system to mount the kernel image in its root file system. Also, at run time, Linux uses space to store system information or hardware device information in proc file system or sysfs of the RAM disk. The traditional RAM disk file system acts as a regular file system that is mounted in the memory device in a single computing system. The RAM-based file systems used in Linux have no durability, which means that if the system’s power turns off, the data of the RAM-based file systems would disappear. Hence, to ensure durability, dumping from RAM disk drive to the hard disk drive should be performed synchronously [14].

The development for RAM-based storage drive has been more revitalized as computing systems require lower latency for single storage I/O operations, especially, applications that use distributed storage systems such as big data databases and cloud computing systems. Distributed RAM storages in cluster environments have been studied [14,17,18]. To overcome the volatility of RAM, RAMcloud [17] provides durability and persistency in a cluster environment, where each node uses RAM as the main storage. Every node replicates each object in the RAM storage and responds to write requests after updating all the replicas. Hence, reliability is ensured even if a node fails. Additionally, modified data are logged to two or more nodes and the logs are asynchronously transferred to a non-volatile storage to achieve durability.

A Solid-State Hybrid Disk (SSHD) is being used on a personal computer, in which SSHD is composed of SSD and Hard disk drive (HDD) inside the storage device. In the SSHD, Several GB of SSDs significantly improve overall performance. Modern state-of-art storage systems employ tiered storage devices [26,27,28]. For the tiered storage device, the DBR DHRD scheme, proposed in this paper, can replace SSD in SSHD device. It can significantly improve I/O performance with DBR method.

On the other hand, recently, some new memory-oriented devices are released to give better I/O throughput for memory-intensive applications. The memory devices are connected to CPU via PCIe NVMe (non-volatile memory express) interface, and the controller chip manages the hybrid storage of the memory and non-volatile memory SSD. The main operation of the controller is caching other NVMe SSDs connected to the systems for accelerating storage I/Os. It provides fast response time and high throughput for both random as well as sequential reads and writes at block I/O levels. However, it does not provide byte-level addressable I/O in the internal operations as it is connected to the PCIe bridge interface. Moreover, the dedicated hardware device has limitations in terms of using and enhancing internal I/O mechanism at the software level.

The proposed hybrid storage system is different from memory-oriented device and provides advantages such as the use of a legacy storage device without any additional hardware devices to provide fast response time for read requests using the new read scheme, and durability of RAM disk drive for write requests. The proposed read scheme in RAM-based disk driver uses byte addressability of RAM device for fast response time. The concept of byte-addressable I/O was proposed in our prior work [29], but this paper presents a new byte-address read scheme that can be mixed with buffered writes to apply it to the hybrid RAM disk. The read performance can be improved with the help of byte addressability of RAM while providing durability similar to that of non-RAM disk systems.

## 3. Durable Hybrid RAM Disk

The DHRD is a hybrid storage that consists of volatile memory and non-volatile storage, while providing the same durability as that of non-volatile storage. In addition, it improves the read performance of RAM disks by using a new read interface that is different from buffered I/O and direct I/O. Read requests are served by the volatile memory, while the write operations are performed on both the volatile memory and the non-volatile storage simultaneously. Therefore, it provides the same durability as that of a non-volatile storage. Read performance is determined by the volatile memory but write performance depends on the non-volatile storage. The DHRD can be used in areas where read performance is more important than write performance and data durability is mandatory. For example, it can be applied to read intensive in-memory databases for sustainable IoT devices. The detailed operations of the proposed system are explained in the next subsections.

### 3.1. Architecture

Figure 2 shows the software architecture of the DHRD to provide data durability. The DHRD consists of high-performance volatile memory and non-volatile storage. The high-performance volatile storage device can be implemented as a RAM disk (RAM disk software). The non-volatile storage device can be implemented with flash-based SSDs. Non-volatile storage devices are generally slower than volatile memory storage devices but do not lose data during power failure. In the proposed system, the volatile memory storage is used as the main storage area, and the updated data in the volatile memory also gets updated to the non-volatile storage device synchronously.

The DHRD is like a mirrored RAID that consists of a RAM disk and a flash-based SSD, where read requests are served only from the RAM disk and write requests are duplicated to both the RAM disk and the SSD. The RAM disk is mirrored with the SSD. Hence, the RAM disk can be recovered from SSD even if the RAM disk loses its data due to a sudden power failure

When the system restarts, the RAM disk is automatically initialized with the data in the SSD. Depending on the capacity of the RAM disk, it might take a long time to load all the data onto the RAM disk. This paper presents a technique that allows immediate response that takes care of the initialization and hence, continues to serve I/O requests during the long initialization period.

The page cache that is used by block devices exhibits unnecessary memory copy overhead for RAM disks. The proposed solution provides a cache bypassing read like direct I/O. This uses the stringent block-level interface, where the buffer size, buffer address, request size, and request position must be multiples of the logical block size. However, the proposed read scheme used in DHRD provides a byte interface that has no constraints. This new read interface is described in detail in Section 3.4.

### 3.2. Basic Primitives

There are three primitive operations in DHRD, read, write, and initialization. Figure 3 shows these primitive operations of DHRD. Each primitive operation acts as follows:Reads: Figure 3a shows the read operation of DHRD. The data contained in the RAM disk are always the same as those in the SSD; therefore, all read requests delivered to DHRD are forwarded only to the volatile memory. In other words, read operations are performed only in the RAM disk.Writes: All write requests delivered to DHRD are sent to both the RAM disk and the SSD. After two writes are completed in these two lower devices, the response for the request is delivered to the upper level of the DHRD. The endio in the figure represents the response for the request meaning the I/O is completed. As a summary, the write operation works like a mirrored RAID.Initialization: The initialization is performed after the system boots. When the system boots, DHRD copies the contents of the SSD to the RAM disk so that the RAM disk can now become the SSD. Generally, the initialization time is quite long; however, the read and write operations can be performed during this initialization time itself in the DHRD. The detailed read and write operations during the initialization is described in the next subsection.

Data can be completely restored from the SSD despite a power failure because the SSD always keeps up-to-date data. The data recovery efficiency of the DHRD depends on the mounted file system. The read performance is determined by the RAM, but the write performance has a bottleneck in the SSD. Consequently, the DHRD is applicable to a server that requires high durability and higher read performance than write performance.

### 3.3. Rapid Resilience with a Fast Initialization

As soon as a system starts, the RAM disk has no data; however the SSD has valid data in the DHRD, and hence, the RAM disk needs to be filled with the contents of the SSD. The DHRD initializes the RAM disk with the data that are in the SSD so that the RAM disk has the same data as the SSD. It takes a long time for this initialization as it is performed through read sequences from SSD device. The DHRD provides data consistency even if I/O requests are delivered to the DHRD during the copy operations from SSD to RAM disk at initialization. Consequently, it allows rapid resilience with fast boot response. There are two operations during initialization: write and read, and there are several cases for each request. DHRD performs proper policy according to the requests.

#### 3.3.1. Writes During Initialization

Figure 4 shows how write requests are processed during the initialization stage. Data blocks are divided into chunk units. Each chunk consists of multiple sectors. The chunks are sequentially copied from the SSD to the RAM disk. As shown in Figure 4, Chunks 0 to 2 were copied from the SSD to the RAM disk and Chunk 3 is being copied. Chunks 4 to 6 have not been copied yet. Write requests are classified into three cases as follows:Case 1: A write request sent to a chunk before being initialized is blocked until the initialization for that chunk completes. When the DHRD has finished copying the chunk to the RAM, all blocked write requests to the chunk are resumed and processed as the initialized chunk.Case 2: Write requests to initialized chunks are processed as normal writes. This means that the write requests are delivered to both the SSD and the RAM disk.Case 3: A write request to an uninitialized chunk is sent only to the SSD. The data written to the SSD will later be copied to the RAM disk by the initialization process. A write request locks the corresponding chunk and unlocks it after finishing the write operation. When the locked chunk is chosen for initialization, the initialization process is suspended and resumed only when the chunk is unlocked by the completion of the write operation. As shown in Figure 4, while a write request to the uninitialized Chunk 4 is being processed, Chunk 4 is locked, Chunk 3 has finished initialization, and the next initialization for Chunk 4 is blocked. The blocked initialization resumes after all writes for Chunk 4 are completed.

#### 3.3.2. Reads During Initialization

Read processing is classified into two cases as follows:Case 1: Read requests to initialized chunks are processed only on the RAM disk.Case 2: Read requests to chunks that are being initialized or were uninitialized are delivered only to the SSD.

This scheme can improve the boot response of the DHRD system. However, requests may not be processed with the best performance during initialization.

### 3.4. Direct Byte Read

The traditional RAM disk is implemented as a block device that is better suited in the form of disks rather than as RAM disks. The block device causes an additional memory copy from the disk cache, but, on the other hand, the RAM disk does not need this disk cache. Here, the disk cache is integrated with the page cache in the Linux kernel.

The traditional buffered I/O uses the page cache, which degrades the performance of the RAM disk. The traditional direct I/O requires that the request parameters be aligned in the logical block size. We need a new I/O interface that can process byte–range requests without the page cache.

This paper presents a new I/O that is optimized for the DHRD. It can process byte–range read requests that bypasses the page cache and uses the buffered write policy for the SSD. The new I/O requires a modified Virtual File System(VFS) in the Linux operating system and an extended block device interface.

Figure 5 compares redundant memory copy with a direct byte read (DBR). The DHRD without the DBR is presented only as a block device, and performs I/O with the page cache. If the DBR is applied to the DHRD, data can be copied directly from the memory of the RAM disk to the user memory without having to go through the block layer.

#### 3.4.1. Compatible Interface

Applications using buffered I/O can use a DBR without modification. Applications use the conventional buffered I/O interface to use the DBR. For direct I/O, the address of the application buffer memory, size of the application buffer memory, request size, and request position must be aligned in the logical block size. The DBR has no alignment restrictions on request parameters. The DBR processes I/O requests in bytes. There is a requirement for the block devices to provide an additional interface for the DBR, but DBR-enabled block devices are compatible with conventional block devices. Thus, the DBR can use the existing file systems.

The applications use the file position in bytes, the buffer memory in bytes, and the size in bytes for I/O. However, the block device has a block-range interface in which all the parameters are multiples of the logical block size. In the traditional I/O interface, the file system in conjunction with the page cache converts a byte–range request into one or more block-range requests. Thereafter, the converted block-range I/O requests are forwarded to the block device.

The DBR requires a DBR-enabled block device, a DBR-enabled file system, and a DBR module in the Linux kernel. A DBR-enabled block device has the traditional block device interface and an additional function that processes byte–range requests. The DBR-enabled file system also has one additional function for DBR. The DBR-support function in the file system can be simply implemented with the aid of the DBR module.

When the kernel receives an I/O request for a file that is in the DBR-enabled block device, the request is transferred to the DBR function of the DBR-enabled block device through the DBR interface of the file system. Therefore, the byte–range request of the application is passed to the block device without transformation.

#### 3.4.2. Direct Byte Read and Buffered Write

The SSD processes only block-range requests, so the SSD cannot use the new I/O. The SSD is used for write requests in the DHRD, but not for read requests. Therefore, the DHRD processes write requests using the traditional block device interface that involves the page cache, while read requests are processed by the direct byte read (DBR). Figure 5 shows the read path and the write path of the DHRD with the DBR. The DHRD uses a buffered write policy that uses the page cache and DBR, which does not use the page cache. To maintain data integrity when read requests and write requests are delivered to the DHRD simultaneously, the DHRD operates as follows:Page not found: When a read request is transferred to the VFS, the VFS checks whether there is buffered data in the page cache. If it is not there, the read request is processed by the DBR.Page found: If there is a buffered page that corresponds to the read request, the data in the buffered page is transferred to the application buffer.

This scheme provides data integrity even though byte-level direct reads are mixed with traditional buffered writes.

## 4. Evaluation

### 4.1. Experimental Setup

This section describes a system that we build to measure the performance of the DBR, DHRD, and evaluation results of the proposed DHRD in comparison with a legacy system. For the performance evaluation, the proposed DHRD is compared with SSD RAID-0 and a traditional RAM disk. Throughout the section, we will denote the DHRD having DBR capability as ‘DBR DHRD’ to differentiate it from the basic DHRD. Also, we denote the software RAM disk as RAMDisk.

The system in the experiments uses two SSDs and 128 GB of DDR3 SDRAM 133 MHz and dual 3.4 GHz processors that have a total of 16 cores. Although the performance evaluation has been performed on high-end IoT platform equipped with multicore processor, we note that the performance of DHRD and DBR in terms of IO throughput and bandwidth is not affected by the number of CPU cores because most of the internal operations of DHRD and DBR consists of IO bound operations, not CPU bound operations. The SSD RAID is a RAID level 0 array that consists of two SSDs and provides 1.2 GB/s of bandwidth. A Linux kernel (version 3.10.2) ran on this machine hosting benchmark programs, the XFS filesystem, and the proposed DBR DHRD driver. We developed DHRD modules in the Linux kernel and modified the kernel to support DBR. The DHRD consisted of a RAMDisk and a RAID-0 array consisting of two SSDs. The RAMDisk used 122 GB of the main memory.

We did performance evaluation with various types of benchmark programs to show its feasibility with the aspect of various viewpoints regarding sustainability in IoT-based systems. Those benchmark programs can cover several IoT devices such as Direct Attached Storage(DAS), Personal Cloud Storage Device (PCSD), Solid-State Hybrid Device (SSHD), and Digital Video Recorder and Player(DVR), which requires advanced I/O operations.

### 4.2. Block-Level Experiments

The first benchmark evaluations are testing for block-level I/O operations. This test is for storage-oriented devices such as DAS, since DAS uses dense block-level I/O operations. In the block-level benchmark, block-level read and write operations without file system operations are done with the benchmark running, then the throughput of the read and write block I/O operations are measured. The results of block-level benchmark evaluation are plotted in Figure 6, where it plots throughputs of random read and random write workloads at block level.

At first, Figure 6a shows the performance of random reads in the block devices without a file system. In the block-level I/O operations, the block devices could be driven by buffered I/O or direct I/O, so these were applied to the SSD RAID-0, RAMDisk, and DHRD, respectively. The DBR DHRD does not distinguish between buffered I/O and direct I/O for reads, instead always treats them as DBR. As shown in the results, the proposed DBR DHRD showed 64 times better read throughput than ‘SSD RAID-0’, which uses direct I/O. On an average, the write throughput of the DHRD with direct I/O was twice that of the DHRD that used buffered I/O. The DBR DHRD showed 2.8 times better read performance than the DHRD that used direct I/O. DBR is implemented as light weight codes, while direct I/O has more complex computing overhead than DBR that has less locks and has no page cache flush and waiting calls. The DBR DHRD, which has low computing overhead and no redundant memory copy, showed the highest read performance.

The write performance of the DHRD depends on the SSD. As shown in Figure 6b, the write performance of the DHRD and that of the SSD RAID-0 are almost the same, but the write performance of DHRD is 3% lower than that of ‘SSD RAID-0’ because the DHRD includes additional operation in the RAMDisk. The write performance of the RAMDisk is superior to others. However, unlike the RAMDisk, the DHRD and the SSD provide persistency. For DHRD with direct I/O, the performance was about 5 times higher when the number of processes were 32 than when the number was 1. The reason being that the SSD consists of dozens of NAND chips and several channels so that the maximum performance of the SSD can be achieved by several simultaneous I/O requests. The DHRD with buffered I/O has less impact on the degree of concurrent I/O requests. When an application writes data using buffered I/O, the data is copied to the page cache and an immediate response is sent to the application. Therefore, the accumulated pages are concurrently transferred to the final storage device later, so that this I/O parallelism is better for the SSD of the DHRD.

Figure 7 shows evaluation conducted Storage Performance Council (SPC) traces that consist of two I/O traces from online transaction processing (OLTP) applications running at two large financial institutions and three I/O traces from a popular web search engine [30]. We replayed the SPC traces on the DBR DHRD, DHRD, RAMDisk, and SSD RAID-0 at the block level. The DHRD showed 8% slower performance than the RAMDisk. However, DBR DHRD showed 20% better performance than the RAMDisk and 270% better performance than the SSD RAID-0. The DBR DHRD performed best on SPC workloads that had mixed reads and writes.

### 4.3. File-Level Experiments

Data storage of IoT devices is a kind of remote storage device that lets systems store data and other files for sustainable IoT-based services. In this device, file-level I/O throughput is critical to the system to give best responsiveness. This section presents an evaluation that uses file-level benchmark programs. It exhibited more computing overhead than the block-level benchmarks. In the file-level benchmark running, we do sequential read, sequential write, random read, random write, and mixed pattern of random read/write operations at a file system level with XFS file system [31]. For the sequential benchmark running, a single process does file-level read and write operations, while throughput of random read and write are measured as the number of processes increases to make more complex situations. For each pattern running, DBR DHRD, DHRD, RAMDisk, and SSD RAID-0 are compared. The results of these file-level benchmark evaluation are shown in Figure 8, where throughputs of sequential read and write, random read, random write, and mixed random read/write workloads are plotted.

Figure 8a,b evaluate the sequential and random read/write performance with a 16 GB file on an XFS filesystem. Figure 8a shows sequential read and write performance. As shown in the results, the DBR DHRD gives 3.3 times better sequential read performance than the DHRD in terms of the throughput aspect. It is because the DBR DHRD has half of the memory copy overhead and simpler computing complexity than the DHRD. The write performances of the SSD RAID-0, DHRD, and DBR DHRD were almost the same due to the bottleneck of the SSD as shown in Figure 8d. The performance of the RAMDisk was the best. Figure 8b shows the mixed random reads and random writes, where the ratio of reads and writes was 66:34. Most applications showed similar behavior with this I/O ratios. The DBR DHRD outperformed the DHRD by 16% on average. The DBR DHRD showed 15 times better performance than the SSD RAID-0 on average with the same durability.

Filebench is a file system and storage benchmark that can generate a wide variety of workloads [32]. Unlike typical benchmarks, it is flexible and allows an application’s I/O behavior to be specified using its extensive Workload Model Language (WML). In this section, we evaluate them with the predefined file server workloads among various Filebench workloads. The file server workload runs 50 threads simultaneously, and each thread creates an average of 128 KB of files, adds data to the file, or reads a file. We measured throughputs for four system configurations as the number of files varies from 32 k to 512 k.

Figure 9 shows performance results obtained using file server workloads using Filebench. In the figure, the *x*-axis represents the number of files and the *y*-axis represents throughputs of each system running. The file server workload has a 50:50 ratio of reads and writes. As shown in Figure 9, the DBR DHRD showed 28% and 54% better performances than the DHRD and the SSD RAID-0, respectively. As this workload has many writes, the RAMDisk achieved the best performance. Although RAMDisk shows higher throughput than DBR DHRD, the RAMDisk suffers from low durability. Thus, DBR DHRD can be said to show better performance while keeping reasonable durability when RAM and SSD are used together in the computing system.

### 4.4. Hybrid Storage Devices and DVR Applications

The tiered storage is a data storage method or system consisting of two or more storage media types. Generally, the frequently used data are served from the fasted storage media such as SSD, and other cold data are accessed from a low-cost media such as HDD, where the first-tier storage as the fasted media is usually performed as a cache for the lower-tier storage. Therefore, the first-tier storage is also called a cache tier.

One of the emerging storage devices is a tiered storage such as SSHD, which is a traditional spinning hard disk with a small amount of fast solid-state storage. BDR DHRD can be applied to the solid-state storage in a SSHD as shown in Figure 10, BDR DHRD can replace the solid-state storage of SSHD, thereby improving the performance of the solid-state storage of a SSHD. To see if the performance of DBR DHRD is improved in a tired storage, we compared the I/O performance of tiered storage devices with DBR DHRD. In this experiment, SSD, HDD, DHRD, and DBR DHRD were configured in tiered storage devices. Three-tiered storage models, SSD+HDD, DHRD + HDD, and DBR DHRD + HDD are considered.

SSHD can be implemented by the flashcache [33] module in Linux. The flashcache can make a tiered storage with SSD and HDD. DHRD is implemented as a general block device, so a DHRD device can replace the SSD of a flashcache device. By this way, we can make a SSHD that consists of DHRD and HDD.

PC Matic Research said that the average memory size of desktop computers is 1 GB in 2008, and 8 GB in 2018 [34]. We can forecast that the average size of PC memory will be 64 GB in 2028. PC motherboards can support up to 128 GB of memory in 2019. In this experiment, the tiered storage used 8 GB of memory, which can be used in the mid-sized to high-end desktop computers.

The I/O traces used in the experiment were collected from three general users using a personal computer. One is a system administrator user, two are developers, and their daily I/O traces are collected and used as experimental I/O traces. In those tiered systems, I/O traces collected from users were performed and throughput is estimated. During the experiment, it is assumed that 70% and 80% of all I/O traces are allocated to SSD or DHRD, which is considered to be the cache tier in tiered storage system.

The results are plotted in Figure 11, in that Figure 11a compares three types of tiered storage devices, SSHD(SSD+HDD), DHRD+HDD, and DBR DHRD+HDD, when the hit rate is 70%. Figure 11b compares them when the hit rate of the cache tier is 80%. Both the RAM size and the SSD size are 8 GB, which is a typical size of a commercial SSHD. As shown in the figures, throughput of DBR DHRD+HDD and DHRD+HDD-based tiered storage outperforms SSD+HDD-based tiered storage about several times for each I/O traces. DBR DHRD scheme also outperforms DHRD only, which is the advantage of direct byte-level read operations supported by DBR. If we compare hit ratio of the cache tier in the tiered storage, the higher the cache tier hit ratio, the higher throughput we have when DBR DHRD is used. From the figure, we identify that the throughput DBR DHRD for 80% cache tier hit ratio is increased about 14%.

Lastly, we conducted experiments on reading and rewriting video files, which is a kind of experiment applicable to multimedia-oriented IoT applications. In this experiment, 1.8 GB sized video file is read, modified partially, and save it as another file. For each system configuration, i.e., SSD, DHRD, and DBR DHRD, we did those operations three times and measured overall execution time. The results are plotted in Figure 12. As shown in the figure, DBR DHRD and DHRD were 2.26 times faster and 1.74 times faster than SSD, respectively. From the results, we identify that DBR DHRD can be applied to IoT devices that deal with multimedia data.

## 5. Conclusions

RAM disk is a software-based storage device to provide low latency, which is compatible with legacy file system operations. The traditional RAM disk includes the disk cache; however, the fact is that it does not require disk cache. Another way for a block device to bypass disk cache, Direct I/O is used; however, the parameters must be a multiple of the logical block size for Direct I/O, so a byte-level addressable path from application to storage device does not exist.

This paper introduced the DRB DHRD scheme for hybrid storage systems that is composed of RAM disk and SSD. The proposed DBR-enabled DHRD provides a byte–range interface. It is compatible with existing interfaces and can be used with buffered writes. The initialization procedure of DBR-enabled DHRD can reduce the boot time of the storage device, since it allows general I/O requests during the initialization process itself, while other RAMDisk-based storage cannot support general I/Os during the initialization. Experimental evaluation was performed using various benchmarks that are applicable to various IoT-based systems performing dense I/O operations. In workloads where reads and writes were mixed, the DHRD performed 15 times better than the SSD. The DBR also improved the performance of the DHRD by 2.8 times. For the hybrid storage device, DBR DHRD performed 3 to 5 times faster throughputs than SSHD. Also, DBR DHRD can reduce execution times of multimedia file’s read and write processing.

As the next step of this study, we are exploring a more advanced version of DRB DHRD for further features and for performance improvement. A more rigorous comparison of the performance of this DRB DHRD scheme versus others could be an important task to improve the completeness of the proposed system. We set the more rigorous performance evaluations as our further work.

## Figures and Tables

**Figure 1 sensors-20-02159-f001:**
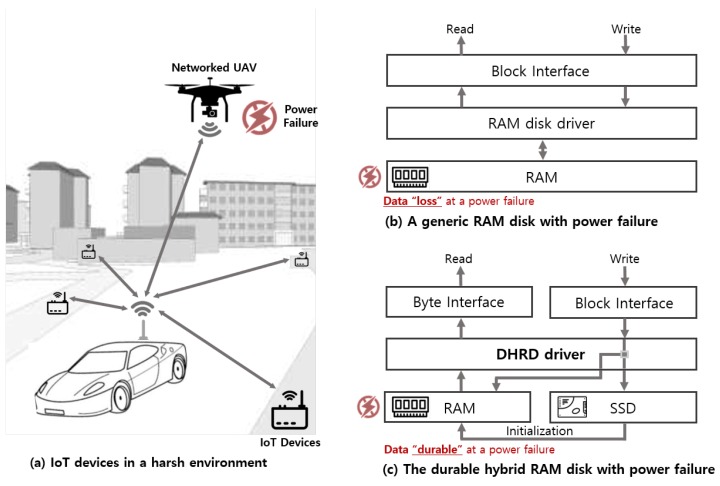
Software stack for the durable hybrid RAM disk (DHRD) with direct byte read (DBR).

**Figure 2 sensors-20-02159-f002:**
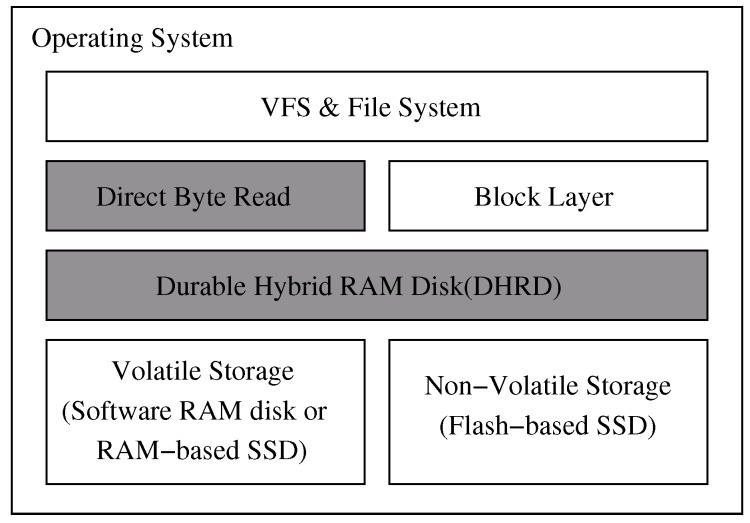
Software stack for the durable hybrid RAM disk (DHRD) with direct byte read(DBR).

**Figure 3 sensors-20-02159-f003:**
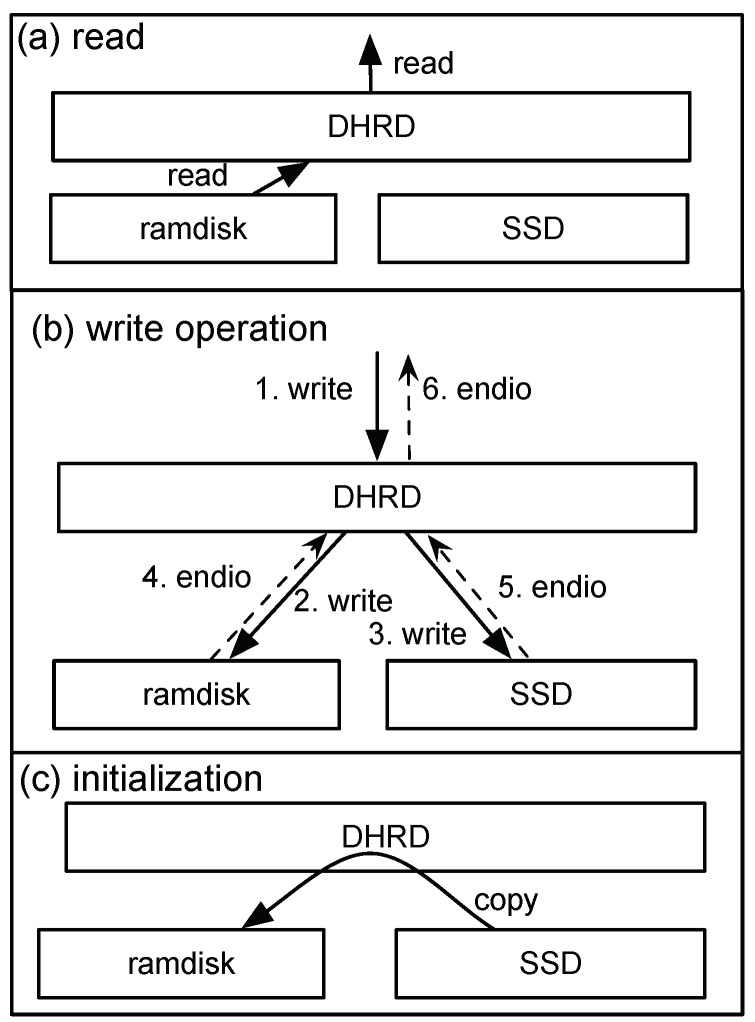
The three primitive operations of the durable hybrid RAM disk; read, write and initialization.

**Figure 4 sensors-20-02159-f004:**
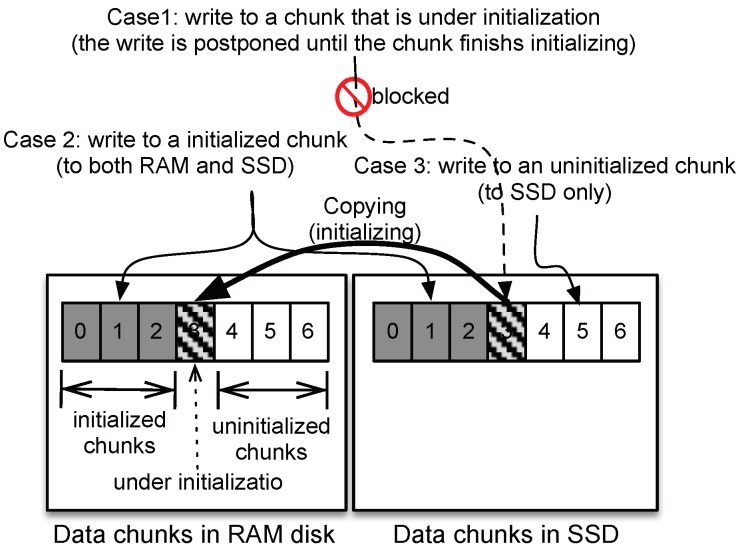
Three write cases during initialization. DHRD ensures data integrity with proper policy for each case.

**Figure 5 sensors-20-02159-f005:**
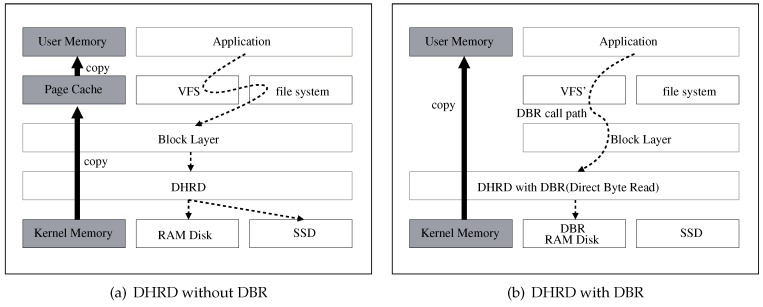
Software stack of DHRD for the cases of redundant memory copy and direct byte read.

**Figure 6 sensors-20-02159-f006:**
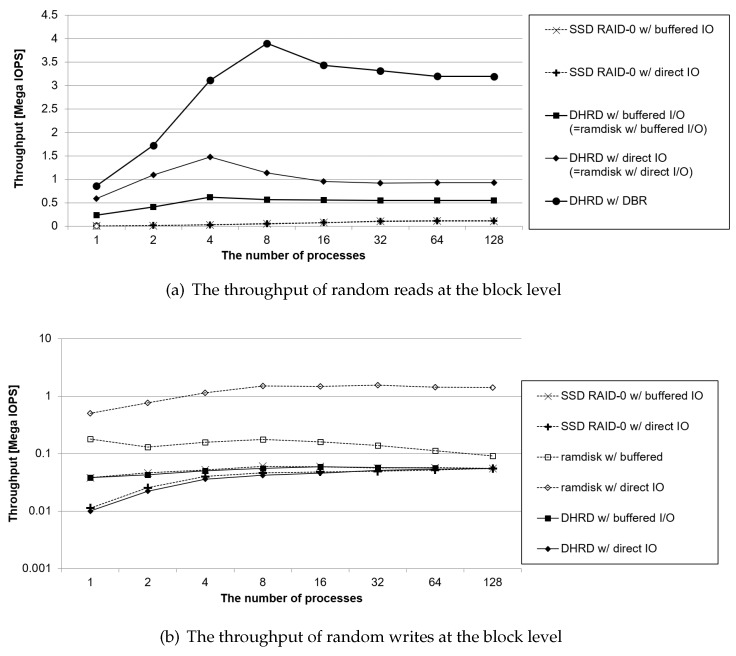
The results of block-level benchmark evaluation. It plots throughputs of random read and random write workloads at block level.

**Figure 7 sensors-20-02159-f007:**
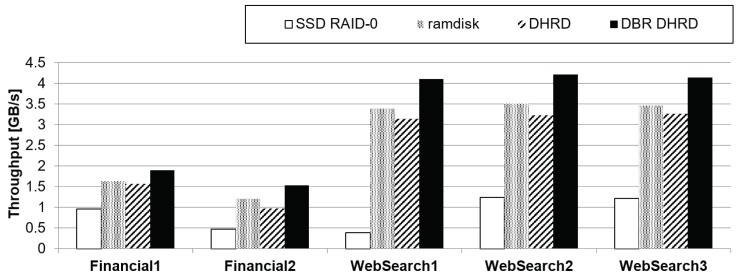
SPC traces: It plots two I/O traces from online transaction processing (OLTP) applications running at two large financial institutions and three I/O traces from a popular search engine.

**Figure 8 sensors-20-02159-f008:**
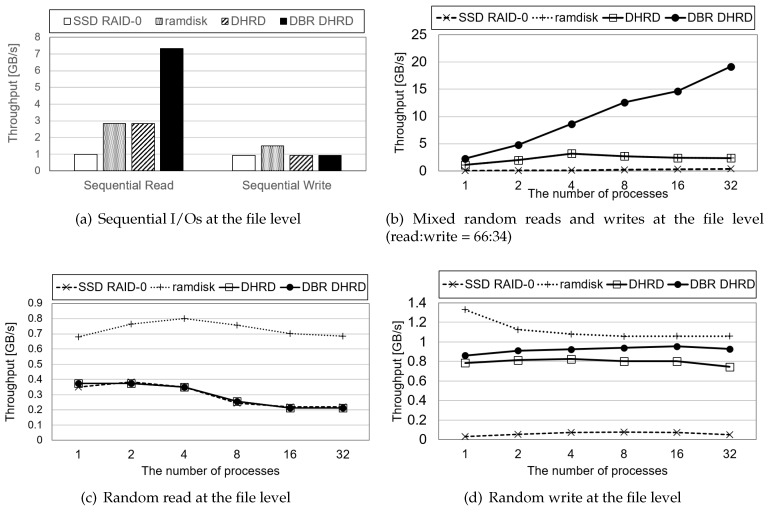
The results of file-level benchmark evaluation. It plots throughputs of sequential I/O, random read, random write, and mixed random read and write workloads at file level.

**Figure 9 sensors-20-02159-f009:**
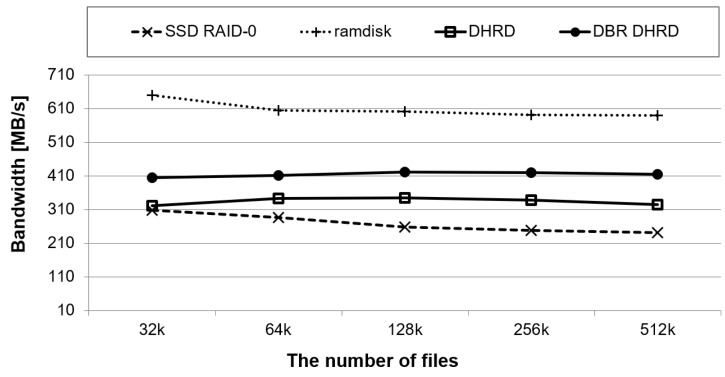
A benchmark using Filebench with fileserver workloads.

**Figure 10 sensors-20-02159-f010:**
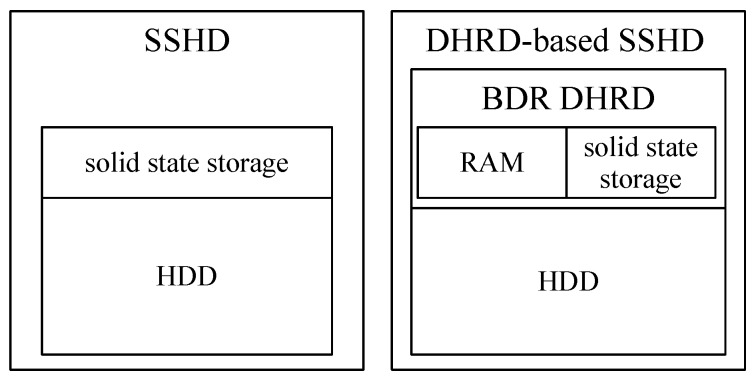
A generic SSHD and a DHRD-based SSHD.

**Figure 11 sensors-20-02159-f011:**
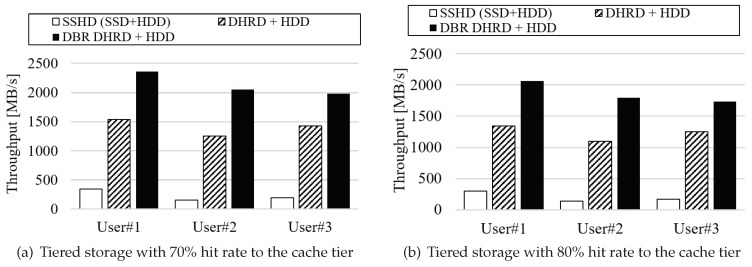
The results of tiered storage in hybrid storage device. It plots I/O throughputs of tiered storage assuming that SSD, DHRD, and DBR DHRD are used as a cache tiered in a tiered storage.

**Figure 12 sensors-20-02159-f012:**
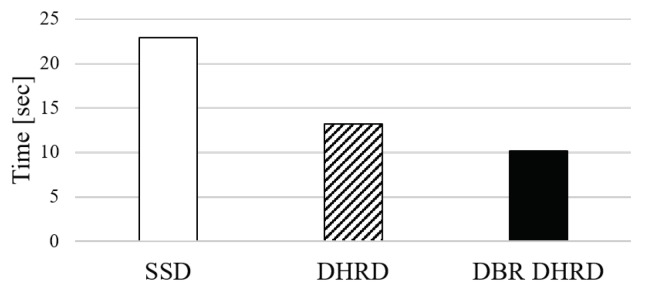
A result of reading and writing for video files.

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
