# Peer review of "A Durable Hybrid RAM Disk with a Rapid Resilience for Sustainable IoT Devices"

_sensors, 2020, doi:10.3390/s20082159_

Round 1

Reviewer 1 Report

The manuscript is about a hybrid RAM disk for IoT. The topic is very interesting for computer architecture venues not for sensor newroks or IoT related topics. 

The method has been reported for 16 cores. A device with such number of cores cannot be classified as an end IoT device or gateway. 

Author Response

Reply Letter

April 1, 2020

Dear Reviewer of this manuscript:

First of all, the authors would like to thank the anonymous reviewer for their time and for their valuable comments. We have very carefully revised the manuscript in accordance with the reviewer' comments. We are now enclosing our response to the reviewer's comments. All of the comments of the reviewer were very helpful for improving the quality of this paper.

Thank you very much.

Sincerely,

Prof. Dr. Sung Hoon Baek (First Author)

Prof. Dr. Ki-Woong Park (Corresponding Author)

Reviewer 2 Report

Your paper describes an interesting proposal. I understand that surely more tests will be necessary to confirm the indicated results, but even so, in my opinion it could be useful to take IoT technology a step further in both speed and data storage security and help other researchers who They are currently working with IoT technology. That is why I believe that the paper should be accepted for publication.

Author Response

(The authors gave the same response as above.)

Reviewer 3 Report

The authors have introduced DRB DHRD scheme for hybrid storage system that is composed of RAM disk and SSD. 

Experimental results presented are very good. However, the relation to IoT devices is not clear in the manuscript, with the exception of a few statements.

I would highly recommend to either change the title, or add more relevance to IoT devices through an extended literature review.

Author Response

(The authors gave the same response as above.)

Reviewer 4 Report

In this paper, the authors proposed a new RAM disk and SSD hybrid storage system, named as DRB DHRD. Experiments show that DRB DHRD has better performance on different IoT devices while providing persistency. The paper is interesting and well-organized. This reviewer has only some small comments:
1) In 4.2, why the ramdisk read performances are not shown in figure6(a)?
2) Figure (8)-(12)’s position needs to be well organized.
3) Will DRB DHRD be open source?

Author Response

(The authors gave the same response as above.)

Reviewer 5 Report

The authors propose a DRB DHRD scheme for hybrid storage system composed of RAM disk and SSD suitable to be used with IoT devices.

The idea of paper is interesting, seems to be a new one, and it is very well explained and tracked. The paper is split into appropriate parts-subparts and the entire content of the paper is easy to track and understand.  

The obtained results are satisfactory and better reflect the quantitative and qualitative aspects of the paper. However, a  more rigorous comparison of the performance of this DRB DHRD scheme versus others could improve the quality of the paper.

Author Response

(The authors gave the same response as above.)
